# MicroRNA miR-181—A Rheostat for TCR Signaling in Thymic Selection and Peripheral T-Cell Function

**DOI:** 10.3390/ijms21176200

**Published:** 2020-08-27

**Authors:** Zoe Grewers, Andreas Krueger

**Affiliations:** Institute for Molecular Medicine, Goethe University Frankfurt am Main, 60590 Frankfurt am Main, Germany; zoe.grewers@kgu.de

**Keywords:** thymus, selection, unconventional T cell, T cell receptor, signaling, microRNA, miR-181

## Abstract

The selection of T cells during intra-thymic d evelopment is crucial to obtain a functional and simultaneously not self-reactive peripheral T cell repertoire. However, selection is a complex process dependent on T cell receptor (TCR) thresholds that remain incompletely understood. In peripheral T cells, activation, clonal expansion, and contraction of the active T cell pool, as well as other processes depend on TCR signal strength. Members of the microRNA (miRNA) miR-181 family have been shown to be dynamically regulated during T cell development as well as dependent on the activation stage of T cells. Indeed, it has been shown that expression of miR-181a leads to the downregulation of multiple phosphatases, implicating miR-181a as ‘‘rheostat’’ of TCR signaling. Consistently, genetic models have revealed an essential role of miR-181a/b-1 for the generation of unconventional T cells as well as a function in tuning TCR sensitivity in peripheral T cells during aging. Here, we review these broad roles of miR-181 family members in T cell function via modulating TCR signal strength.

## 1. Introduction

T cells are at the center of the adaptive immune response. Following stimulation with cognate antigen by professional antigen-presenting cells (APCs), they become activated to either directly fulfill effector function, such as elimination of virus-infected cells, or elicit and enhance the function of other cell types, including B cells and cells of the innate immune system. In addition to these functions executed by so-called conventional T cells (Tconv), multiple subsets of unconventional T cells exist. Among those are regulatory T (Treg) cells, capable of suppressing unwanted immune responses to self or innocuous antigens, as well as invariant natural killer T (iNKT) cells, mucosal-associated invariant T (MAIT) cells, and γδT cells, which are characterized by innate-like functions.

The T cell receptor (TCR) forms the central unit of antigen recognition by T cells. The majority of T cells, including virtually all Tconv cells, express αβTCRs. These TCRs, with the exception of some TCRs of unconventional T cells, recognize peptide antigens in the context of MHC molecules (pMHC). A smaller population of T cells express γδTCRs. The α, β, γ, and δ chains of TCRs are generated through somatic recombination of *Tra*, *Trb*, *Trg*, and *Trd* genes, respectively, during T cell development in the thymus. Somatic recombination is essentially random, thus generating a broad array of TCRs, many of which are incapable of recognizing pMHC antigen and the others not discriminating between self and non-self-antigens. Therefore, in order to create a pool of T cells capable of recognizing foreign antigen, but at the same time not eliciting autoimmune disease, non-functional and autoreactive cells must be removed from the T cell pool. These processes are termed positive and negative selection, respectively, and are both controlled by signaling through the TCR. Accordingly, rather than mediating digital on/off signals, in thymocytes the TCR has to integrate signals inducing multiple possible fates: T cells that do not possess a detectable affinity to pMHC die by neglect, those that have low to intermediate affinity to pMHC are positively selected and the fraction of cells that bind pMHC with high affinity is deleted or driven into a pathway called agonist selection to clonally divert thymocytes into unconventional T cell lineages (Figure 1) (for review see [1,2]).

The TCR consists of a disulfide bridge linked αβTCR or γδTCR dimer, complexed with CD3γ, CD3δ, CD3ε, and the ζ chain. Signaling is initiated by phosphorylation of immunoreceptor tyrosine-based activation motifs (ITAMs) within the TCR:CD3 complex by the Src family kinase Lck. Phosphorylation of ITAMs results in recruitment of the kinase ZAP-70, which in turn results in the formation of adaptor scaffolds and the activation of multiple downstream signaling pathways including Ca^2+^ signaling, MAP kinases, such as Erk, PI3K/Akt signaling, as well as NF-κB. Given the central role of phosphorylation, in particular during proximal TCR signaling, it is unsurprising that phosphatases play a critical regulatory role (for review see [3]). Notably, phosphatases may act as positive or negative regulators of the TCR signals by removing inhibitory or activating phosphates, respectively. The receptor protein tyrosine phosphatase CD45 constitutes a key positive regulator as it dephosphorylates inhibitory phospho-tyrosines of Lck. However, it may also act as a negative regulator through dephosphorylation of CD3ζ, suggesting that it contributes to tuning of TCR signals in response to antigen [4]. Calcineurin, in response to increased Ca^2+^ levels, dephosphorylates, and thereby induces nuclear translocation of NFAT transcription factors [5]. Conversely, Ptpn22 and SHP-1 (Ptpn6) constitute critical negative regulators [6]. In particular, SHP-1 has been implicated as a central component of a negative feedback loop allowing for pMHC-specific T cell activation at low ligand concentrations in the presence of an excess of low-affinity ligands [7,8]. In this model, SHP-1 forms a critical rheostat for a digital on/off response at the level of Erk phosphorylation. Another group of phosphatases linked to Erk activation are dual-specificity phosphatases (DUSPs) capable of dephosphorylating MAP kinases at both serine/threonine and tyrosine residues [9].

Quantitative as well as qualitative differences elicited by positively and negatively selecting pMHC ligands have been described. Clonal deletion is induced rapidly within a few hours after signaling [10]. Consistently, in an inducible model of ZAP-70 activation, ZAP-70 activity of 1 h was sufficient to induce negative selection [11]. Conversely, sustained ZAP-70 activity of at least 36 h was required to promote positive selection [11]. Similarly, sustained activity of Erk is required for positive selection, whereas Erk activity rapidly wanes after induction of negative selection [12,13,14]. Moreover, whereas negatively selecting pMHC ligands restrict Erk signaling to the plasma membrane, positively selecting ligands induce distribution of the Erk signaling module throughout the cytoplasm [15]. The role of phosphatases during thymic selection has not been comprehensively characterized. DUSP5 and DUSP6 (also termed MKP-3) have been implicated in positive selection. Thus, transgenic expression of DUSP5 blocks T cell development at the CD4/CD8 double-positive (DP) stage, the stage where thymic selection is initiated, whereas expression of a dominant-negative mutant of DUSP6 promotes positive selection in a dose-dependent manner [16,17]. Like in mature T cells, SHP-1 has been suggested to modulate TCR signaling thresholds to discriminate between positive and negative selection. Conditional deletion of SHP-1 in thymocytes revealed moderate effects on TCR signal strength [18]. Modulation of SHP-1 activity has been proposed to be the key mechanism of action of Themis, a modulator of TCR signal strength predominantly expressed in DP thymocytes [19,20]. However, it remains controversial whether Themis inhibits or promotes SHP-1 activity [19,21]. Taken together, in thymocytes, the TCR converts signals elicited by pMHC ligands with a continuum of affinities to a digital outcome of positive selection or clonal deletion. Agonist selection, during which unconventional T cells are positively selected by high-affinity ligands will be discussed below.

## 2. The miR-181 Family

MiRNAs constitute a group of noncoding RNAs of 18–22 nucleotides in length and predominantly function as post-transcriptional repressors (for review see [22]). Following a series of processing steps by endonuclease complexes, mature miRNAs are incorporated into the RNA-induced silencing complex (RISC). The miRNA-loaded RISC represses mRNA through two potential mechanisms, translational inhibition or decay of the mRNA [23]. The rules governing the choice of repressive mechanism remain unclear, but both processes may also be kinetically linked [24]. A six to eight nucleotide so-called seed sequence at the 5′ end of the miRNA determines binding to mRNA targets [25]. Accordingly, each individual miRNA is capable of binding a broad array of mRNAs. Although hierarchies of target choice have been well established biochemically, cell-type-specific context is likely to contribute to miRNA-mediated targeting in vivo [26,27,28]. A number of miRNAs, including miR-17~92, miR-142, miR-146, miR-148, miR-150, miR-155, and others, have been implicated in the development and function of the immune system and T cells in particular (for review see [29,30,31]. Here, we focus on the miR-181 family and discuss its role in the development and function of T cells and as a rheostat of TCR signaling.

The evolutionary conserved miR-181 family consists of 6 members, genetically clustered into pairs, miR-181a/b-1 (on chromosome 1 in mice and humans), miR-181a/b-2 (on chromosome 2 in mice and 9 in humans), and miR-181c/d (on chromosome 8 in mice and 19 in humans) (Figure 2A). Mice lacking two of the three clusters have reduced body size and shortened overall survival, whereas deficiency of all three clusters appears to be incompatible with life, indicating that the miR-181 family exerts vital functions [32]. Accordingly, it has been suggested that miR-181 family members have roles in embryonic development, as well as cardiovascular and nervous systems (for review see [33].) In the T lineage, miR-181a/b-1 are the most prominently expressed family members, whereas lower levels of miR-181c/d and virtually no miR-181a/b-2 are detectable [34,35]. Despite low expression levels, minor roles in T-cell function and development were ascribed to miR-181c/d [36,37]. However, these roles were mostly deduced from overexpression experiments, which are difficult to interpret as all miR-181 family members share the same seed sequence. Consistently, specific deletion of miR-181d alone did not reveal an opposing phenotype when compared to its transgenic overexpression [36]. Expression of miR-181a/b-1 is dynamically regulated (Figure 2B). (There is currently no evidence, that miR-181a-1 and miR-181b-1, the two miRNAs of the miR-181a/b-1 cluster are differentially regulated, although miR-181a levels appear to be consistently higher [34,35]. In addition, both members share the same seed sequence, and their differences in the sequence of the more stable 5p strand are restricted to positions 9, 10, and 21, suggesting a substantial overlap in putative targets. Therefore, targeted deletion of miR-181a/b-1 is likely to represent the most physiological loss-of-function model, whereas ectopic expression of miR-181a might, to a considerable degree, also affect targets of other miR-181 family members. Selective inhibition of miR-181a by antimiRs may be difficult to interpret because the effect of these antimiRs on silencing other miR-181 family members is difficult to control. This should be kept in mind for the description of experimental findings below.) MiRNA sequencing in thymocytes showed a steep increase in relative miR-181a/b expression beginning at the CD4/CD8 double-negative (DN)3 stage of development and reaching a peak in DP thymocytes [35]. In fact, this study suggested that miR-181 constituted almost half of all miRNAs in this thymocyte population. Other studies have reported a similar dynamic regulation, also reporting absolute copy numbers, which are slightly reduced in DP thymocytes due to their small size [38,39]. In peripheral T cells, levels of miR-181a are even further reduced when compared to their intrathymic counterparts and expression is even further reduced in Treg cells [40]. Both CD4 and CD8 T cells loose expression of miR-181a upon activation in vitro, consistent with lower expression of miR-181a in memory vs. naive T cells [41,42]. Moreover, miR-181a is dynamically expressed in T cells over the lifetime of an organism, with its levels progressively decreasing with increasing age [43,44]. Together, dynamic regulation suggested that miR-181a functions predominantly in the thymus.

A TCR signal modulatory function of miR-181a was first suggested by Li and colleagues in 2007 [39]. An ectopic increase of miR-181a expression in mature T cells resulted in increased sensitivity to peptide antigens as well as increased basal phosphorylation levels of Lck and Erk. Likewise, the inhibition of miR-181a via antagomirs reduced TCR sensitivity. Consistent with its role in modulation of TCR signaling multiple phosphatases involved in T cell receptor signaling such as SHP-2, PTPN22, DUSP5, or DUSP6, were identified as targets of miR-181a (Figure 2C) [39]. Although SHP-1 was not directly repressed, ectopic expression of miR-181a interfered with SHP-1 forming a physical interaction with Lck, suggesting that miR-181a might also indirectly modulate TCR signaling via SHP-1. Notably, in this study, repression of any of the individually targeted phosphatases was insufficient to mimic the effect of ectopic expression of miR-181a on TCR signaling. Thus, miR-181a may be considered a paradigmatic example of miRNA co-targeting networks [45]. Given that expression of an individual miRNA frequently only results in modest repression of a single target mRNA, co-targeting networks have been proposed to have evolved to permit a robust, yet tunable, impact of an individual miRNA on a given biological process.

## 3. miR-181 in Selection of Conventional T Cells

Initial evidence for a role of miR-181a in thymic selection was obtained in vitro employing fetal thymic organ cultures in conjunction with the application of selecting peptides and synthetic miR-181a inhibitors [39,46]. These experiments indicated that inhibition of miR-181a impaired both positive and negative selection, in keeping with the hypothesis that a reduction in TCR signal strength precludes signaling from low-affinity ligands, thus preventing positive selection, and limits clonal deletion by delivering weaker signals in response to high-affinity peptides (Figure 3A) [39]. Indeed, in a similar experimental setup, a moderate increase in autoreactivity was detected [46]. However, although DP thymocytes were less responsive to TCR triggering, at the level of total thymocytes no overt defects in positive or negative selection in mice lacking miR-181a/b-1 were detected [40,47]. Moreover, these mice did not display any signs of autoimmunity. Rather, miR-181a/b-1-deficient mice were comparatively resistant to the induction of experimental autoimmune encephalitis (EAE) [47]. In contrast to the in vitro experiments described above, loss of miR-181a/b-1 did not result in an increase in the number of autoreactive T cells, although their reactivity to antigen upon immunization was somewhat increased. TCR transgenic mice were employed to assess the role of miR-181a/b-1 in the selection of T cells with fixed affinity. In HY-TCR (recognizing the Y-chromosome encoded SMCY antigen in the context of MHC-I) transgenic male mice, deletion was independent of miR-181a/b-1. However, female mice displayed increased numbers of positively selected mature CD8 single-positive cells [47], indicating either enhanced positive selection or rescue from inadvertent negative selection occurring in female mice. On one hand, enhanced positive selection in the absence of miR-181a/b-1 runs counter to the hypothesis of reduced sensitivity to antigen. On the other hand, there is no clear evidence for negative selection of HY-TCR transgenic cells in female mice. Some evidence for a defect in clonal deletion was derived from bone-marrow chimeric RIPmOVA mice expressing ovalbumin in thymus and pancreas, which carried donor cells expressing the MHC-II restricted ovalbumin-specific OT-II TCR [48]. In these mice clonal deletion of OT-II TCR transgenic cells was impaired in the absence of miR-181a/b-1. Together, these studies fail to present a clear model how miR-181a/b-1 sets TCR signaling thresholds for positive and/or negative selection. Analysis of polyclonal models are limited by the fact that, in these models, shifts in TCR sensitivity are difficult to detect. Conversely, models employing fixed TCRs might be limited by the basal affinity for self-antigens, which is intermediate for the OT-II TCR, but low for HY. Moreover, as miR-181a/b-1 is predominantly expressed in DP thymocytes followed by a steep decline, differences between TCRs reactive to antigens expressed in the thymic cortex and the medulla might be expected. In addition, based on the studies described above, it cannot be excluded that there are fundamental differences between MHC-I and MHC-II restricted TCRs.

## 4. miR-181 in Unconventional T Cells

So-called unconventional T cells comprise a diverse set of populations that are generated in the thymus without undergoing prototypical positive and negative selection in response to pMHC ligands [31,49]. Such cells may be directly derived from DN thymocytes without going through the DP stage of development. This type of unconventional cells comprises γδT cells as well as some intestinal intraepithelial lymphocytes (iIELs) and iNKT cells. MAIT cells and iNKT cells express semi-invariant αβTCRs and recognize derivatives of vitamin B in the context of the MHC-I-like molecule MR1 and glycolipids in the context of CD1d, respectively [50,51,52]. Both cell types are generated from DP thymocytes and display characteristics of innate lymphocytes having the capacity of secreting effector cytokines without prior stimulation by cognate antigen. Although being selected on classical pMHC ligands, Treg cells, like iNKT cells, require distinctly stronger TCR signals for development when compared to conventional T cells and are therefore considered to undergo agonist selection (Figure 1) [53].

### 4.1. Treg Cells

Treg cells constitute key mediators of peripheral tolerance and are characterized by expression of the lineage-determining transcription factor Foxp3 (for review see [54]). They are generated from two distinct sources. Thymic (t)Treg cells are formed in the thymus through agonist selection, whereas induced (i)Treg cells are generated in the periphery from naïve T cells in the presence of cytokines, such as TGFβ. In the thymus, mature CD25^+^Foxp3^+^ tTreg cells are produced in a two-step process via two possible intermediates: CD25^+^Foxp3^−^ or CD25^−^Foxp3^+^ [55,56,57,58]. Strong TCR signals are required to produce CD25^+^ precursors followed by IL-2-driven maturation [53,59]. The formation of Foxp3^+^ precursors is also driven by, albeit somewhat weaker, strong TCR signals and simultaneously depends on cytokine-driven signals, such as via IL-15 [59]. Unsurprisingly, tTreg cells have distinct TCR repertoires depending on their precursor history and are autoreactive in nature [56]. How Treg cells circumvent clonal deletion and are directed to an alternative pathway termed clonal diversion remains an open question [60]. It has been proposed that Treg cells need strong TCR signals, albeit just below the border to clonal deletion [53]. In addition, a reduction of MHC class II levels on medullary thymic epithelial cells decreased negative selection and resulted in an increased emergence of Treg cells [61]. Protective effects against relatively high TCR signals such as TGFβ signaling have been proposed as well [62]. Despite normal numbers of Treg cells in the periphery of miR-181a/b-1-deficient mice, de novo generation of Treg cells was inefficient in these mice [48]. Notably, loss of miR-181a/b-1 more prominently affected Foxp3^+^ precursors, whereas CD25^+^ precursors were even enriched, but were apparently unable to contribute significantly to the pool of mature Treg cells. As the latter precursors are characterized by stronger TCR signals, it might be assumed that loss of miR-181a/b-1 in these cells in fact induced clonal diversion from negative selection. Indeed, clonal diversion was evident in OT-II TCR transgenic RIPmOVA bone marrow chimeric mice. In contrast, defects in generation of Foxp3^+^ precursors may result from a failure to elicit sufficiently strong signals. In order to support a mechanistic link between Treg cell differentiation, TCR signal strength, and miR-181a/b-1, the Nur77 family member Nr4a2 was inducibly expressed to mimic strong TCR signals [63]. Consistent with the hypothesis that altered TCR signaling thresholds accounted for defective Treg cell development, induction of Nr4a2 restored de novo Treg cell formation in miR-181a/b-1-deficient mice. In the periphery of miR-181a/b-1-deficient mice, homeostatic expansion resulted in the formation of a Treg cell pool with limited TCR diversity and, surprisingly, elevated suppressive capacity, presumably due to elevated levels of the key co-inhibitory receptor CTLA-4 [48]. The underlying mechanism of elevated CTLA-4 expression in miR-181a/b-1-deficient Treg cells remains poorly understood. However, increased suppressive capacity of Treg cells might at least in part explain the absence of overt autoimmunity in these mice as well as their partial resistance to EAE [40,47]. Expression of miR-181a is low in peripheral T cells from healthy mice. However, in the context of type 1 diabetes, elevated levels of miR-181a were associated with impaired tolerance induction as well as autoimmune activation. Of note, increased levels of miR-181a were associated with altered TCR signal strength as well as increased NFAT5 expression and defective induction of iTreg cells. Consistently, iTreg formation was improved in NFAT5 knockout mice but in these mice lost its dependence on miR-181a activity [64].

### 4.2. iNKT Cells and MAIT Cells

Loss of the vast majority of iNKT and MAIT cells constitutes the most prominent T-cell associated phenotype of miR-181a/b-1-deficient mice [32,40,65]. In the thymus, iNKT cells are generated from DP thymocytes through a series of developmental intermediates termed stage 0 to stage 3 and undergo functional differentiation into effector cell types termed NKT1, NKT2 and NKT17, depending on their cytokine expression profile [66]. At stage 0, iNKT cell precursors receive strong TCR signals in conjunction with co-stimulatory signals through SLAM family receptors [53,67,68]. Recently, it has been proposed that these co-stimulatory signals in fact provide negative feedback to prevent TCR-induced clonal deletion [69]. In the absence of miR-181a/b-1, the development of iNKT cells is arrested at stage 0, consistent with the hypothesis that alterations in TCR signaling accounted for this developmental arrest. This hypothesis was further substantiated by the observation of an altered TCR repertoire expressed on iNKT cells escaping the developmental block. Moreover, ectopic administration of an agonist glycolipid ligand as well as transgenic overexpression of the invariant iNKT cell-specific TCRα chain rescued iNKT cell production in the absence of miR-181a/b-1 [40,70]. Of note, loss of prototypical iNKT cells in the liver was partially compensated for by NKT-like cells expressing a γδTCR [71]. Effector cell differentiation of iNKT cells depends on differential TCR signal strength as well [72]. Thus, NKT2 and NKT17 cells were characterized by higher expression levels of the transcription factor Egr2, which serves as a surrogate measure for TCR signal strength. Moreover, mice expressing a partial loss-of-function mutant of ZAP-70 had a selective defect in the generation of NKT2 and NKT17, but not NKT1 cells. Surprisingly, the opposite skewing of NKT effector cell populations was observed in miR-181a/b-1-deficient mice, suggesting additional roles of miR-181a/b-1 at later stages of iNKT cell development [70]. However, far fewer iNKT cells escape the developmental block elicited by miR-181a/b-1 deficiency when compared to ZAP-70 mutation, raising the possibility that these cells do not reflect true NKT cell effector subsets.

Despite recognizing a fundamentally different class of antigens, MAIT cells are highly similar to iNKT cells. In fact, recent transcriptomics studies revealed that, within a given tissue, effector MAIT1 and MAIT17 cells were more similar to NKT1 and NKT17 cells, respectively, than MAIT1 and MAIT17 to their MAIT-cell counterparts across tissues [73]. MAIT cells expressing type 2 cytokines, such as IL-13, have only recently been identified [74]. Due to their paucity in mice, molecular cues for the development of MAIT cells have only recently been identified [75]. Similar to iNKT cells, MAIT cells are derived from DP cells, and for the most part are selected on thymocytes rather than epithelial cells and depend to a large extent on additional signaling from SLAM family receptors. Recently, a subset of MAIT cells has been characterized that is selected on thymic epithelial cells and displays characteristics of naive T cells rather than being pre-activated innate-like cells [76]. Based on evidence obtained from germ-free mice, it has been suggested that MAIT cells are selected on exogenous ligands of microbial origin [75,77,78]. Indeed, a topically administered antigen was shown to be rapidly transported to and presented on DP thymocytes inducing positive selection of MAIT cells [77]. MAIT cell development is also critically dependent on miRNA [75]. Similar to the defect observed upon global deletion of miRNAs, in the absence of miR-181a/b-1 alone, MAIT cell formation was almost fully arrested at developmental stage 1 [65]. Interestingly, despite a near-complete loss of MAIT effector cell differentiation in the thymus of miR-181a/b-1-deficient mice, few mature MAIT cells with a bias toward MAIT1 were detectable in the periphery. This observation suggests that MAIT cells can undergo terminal maturation into effector subsets in the periphery. Ectopic expression of the invariant MAIT cell TCRα chain rescued development of MR1-restricted cells in the absence of miR-181a/b-1, providing indirect evidence for a role of TCR signaling in the mechanism underlying miR-181a/b-1-mediated control of MAIT cell development. Of note, TCR signal strength requirements in MAIT cell formation remain uncharacterized to date, and the question of whether MAIT cells represent agonist selected cells, such as iNKT or Treg cells remains to be answered. 

## 5. miR-181 Function in Peripheral T Cells

Peripheral T cells display critically different outcomes to TCR stimulation when compared to thymocytes. Whereas the latter respond to low-affinity antigens by undergoing positive selection and to high affinity antigens by clonal deletion, the former remain largely unresponsive to low-affinity antigens and become fully activated in response to high-affinity antigens [79,80,81]. These alterations in sensitivity to respond to TCR signals are consistent with lower expression of miR-181 in peripheral T cells, although a causal role for miR-181 in the transition from thymic to peripheral TCR sensitivity remains to be established [39,48].

With the increasing age of an organism, the immune system undergoes a number of fundamental changes (for review see [82,83]). Such changes result in increased vulnerability to infections as well as the limited success of vaccinations. The T cell compartment undergoes substantial alterations during aging. First, the thymus is subject to atrophy, reducing the output of naive T cells already during childhood. Accordingly, in adult humans, homeostatic proliferation is the main mechanism to maintain a naive T cell pool. During aging, homeostatic proliferation declines to a certain extent but is retained over extensive periods of time [84]. Consistently, TCR diversity is somewhat reduced, but presumably not to an extent causing substantial “holes” in the TCR repertoire [84]. Alterations in the composition of T cell subsets may also result in limited T cell responses in elderly individuals. Notably, upon cytomegalovirus (CMV) infection, CMV-specific T cell clones may expand excessively, resulting in “holes” in the memory T cell repertoire, called memory inflation [85]. Moreover, during aging so-called “virtual memory” T cells may expand, which results in curtailed T memory responses [86]. The functional consequences of each of the alterations remain only partially understood. Age-related alterations in cell-intrinsic T cell function were also reported. Thus, long-lived naive CD4 T cells produced less IL-2 in mice [87], and in humans, naive CD4 T cells from aged individuals were less responsive to TCR stimulation and displayed reduced MAP kinase activity [44,88]. Conversely, expression of DUSP6 was elevated in naive CD4 T cells from elderly individuals, whereas the expression of miR-181a was decreased, suggesting regulation of TCR-mediated Erk signaling via miR-181a and its target DUSP6. Both, ectopic expression of miR-181a and silencing of DUSP6 increased responsiveness of CD4 T cells to TCR triggering, whereas inhibition of miR-181a reduced T cell activation [44]. Notably, other typical phosphatase targets of miR-181a, such as Ptpn22 and SHP-2, remained unaffected in elderly individuals, pointing toward context-dependent targeting. Consistent with progressively decreased expression of miR-181a with increasing age, miR-181a levels were elevated in cord blood-derived CD4 T cells, coinciding with moderately enhanced TCR signaling [43]. The functional consequences of dynamic regulation of miR-181a in peripheral T cells were assessed in a mouse model with conditional miR-181a/b-1 deficiency [89]. In the absence of miR-181a/b-1, clearance of lymphocytic choriomeningitis virus (LCMV) was delayed due to inefficient differentiation and expansion of effector T cells. Although memory CD8 T cells were generated, similarly recall responses by these cells were defective. Notably, the effect on CD4 T cells in this experimental system was complex. Presumably due to delayed viral clearance in the absence of miR-181a/b-1, an expansion of CD4 T cells was observed, which masked a cell-intrinsic defect apparent in a competitive setting. Observations in young and old patients infected with West Nile Virus were consistent with these findings. TCR repertoire analysis showed selection toward higher antigen sensitivity and a bias toward Tfh cell differentiation, both consistent with a mechanistic link between TCR signaling and miR-181a/b-1 function in peripheral T cells. Consistently, loss of miR-181a/b-1 in peripheral T cells resulted in upregulation of DUSP6 and limited phosphorylation of Erk [89]. Interestingly, another study reported improved viral clearance in a model of murid gammaherpesvirus 4 infection and a concomitant increase in activated CD8 T cells [90]. The reason for this apparent discrepancy remains unclear. However, the latter study employed pan-miR-181a/b-1-deficient mice, raising the possibility of indirect effects. Taken together, these studies suggest that despite comparatively low expression levels in peripheral T cells, which are further reduced upon activation, miR-181a/b-1 plays a substantial role in controlling viral infection, effector T cell expansion, and memory T cell function.

## 6. Non-TCR Signaling Related Targets of miR-181

Here, we have described the impact of miR-181 on T cell development and function via tuning of TCR sensitivity. These findings are consistent with miR-181 being upstream of a co-targeting network consisting of multiple phosphatases, which act as negative regulators of TCR signaling [39,45]. However, given the property of miRNAs to target a large number of distinct mRNAs, it is unlikely that this co-targeting network constitutes an exclusive means of miR-181 function. In fact, more than 1100 mRNA targets have been predicted for miR-181 in silico, and some of these have also been characterized in the context of T cell development and function. The phosphatase PTEN controls PI3K signaling, which, in turn, induces anabolic metabolism in immune cells. In the case of iNKT cells, miR-181a/b-1 has been suggested to be a non-redundant determinant of metabolic fitness by targeting PTEN [32]. Heterozygous deletion of PTEN partially rescued the defect in iNKT cell development in miR-181a/b-1 deficient mice. However, two other studies failed to detect significant upregulation of PTEN in thymocytes from miR-181a/b-1 deficient mice [47,70]. Although these findings do not formally exclude a role of miR-181a in controlling metabolic fitness, it is not certain whether this is a primary or secondary effect of inadequate TCR-signaling. 

Deletion of miR-181a/b-1 revealed a modest defect in early T cell development as well as increased survival of mice in a model of T cell acute lymphoblastic leukemia [91]. This study suggested an interplay of miR-181-mediated modulation of TCR signaling with restriction of Notch signaling by targeting its inhibitor Nrarp [91].

Alterations in thymic selection can only partially explain the relative resistance of miR-181a/b-1-deficient mice to EAE [47]. Whereas increased suppressive capacity of Treg cells may contribute to this phenotype, sphingosine-1-phosphate receptor 1 (S1PR1) has been proposed as an additional target of miR-181a/b-1 [47,48]. Elevated S1PR1 levels resulted in aberrant migration of miR-181a/b-1-deficient T cells.

Altered expression levels of miR-181a in conventional T cells were shown to be critical in a mouse model of acute graft-versus-host-disease (aGvHD) [41]. Bone marrow transplant (BMT)-recipient mice receiving donor T cells with enhanced miR-181a expression showed no signs of aGvHD and survived for the time of follow-up. However, T cells lacking miR-181a/b-1, exhibited accelerated aGvHD. Of note, expansion of activated T cells with enhanced miR-181a expression was reduced in vitro and in vivo, coinciding with repression of anti-apoptotic Bcl-2 family members Bcl-2, Bcl-x_L_, and Mcl-1 [41]. The relative contribution of these changes in expression versus a potentially increased susceptibility of T cells to TCR-mediated activation-induced cell death remains to be determined.

Loss of miR-181a/b-1 results in delayed clearance of LCMV after infection due to a limited CD8 T cell response [89]. Conversely, miR-181a/b-1-deficiency improved the clearance of murid gammaherpesvirus 4 [90]. The latter was attributed to elevated levels of IFNγ produced in the absence of miR-181a/b-1, presumably via derepression of Id2. Thus, it is possible that, depending on the type of infection, targeting of modulators of TCR signaling or targeting of the IFNγ pathway determines the phenotype of abrogation of miR-181a/b-1 in peripheral T cells.

## 7. Conclusions

Accumulating evidence indicates that miR-181a/b-1 predominantly acts as a rheostat of TCR signaling in both thymocytes and peripheral T cells, most likely by interfering with TCR signal strength via a co-targeting network of negative regulatory phosphatases. Consistent with its dynamic expression profile, the consequences of loss of miR-181a/b-1 are generally more severe during intrathymic T cell development, in particular of agonist-selected T cell populations. Nevertheless, a number of critical questions remain. Although data suggest that the progressive severity in phenotype from development of conventional T cells via Treg cells to semi-invariant iNKT and MAIT cells can be attributed to increasingly narrow requirements in TCR signal strength, this hypothesis is by no means confirmed (Figure 3). For instance, it remains to be formally demonstrated that MAIT cells, like iNKT cells, undergo agonist selection [92]. In addition, agonist selection requires co-stimulatory signals to promote cell survival. Thus, a different type of signal integration might account for the more stringent requirement for miR-181a/b-1 in this process. In addition, not only quantitatively different, but also qualitatively different signals determine the outcome of thymic selection. Therefore, hypothesizing a mere shift in sensitivity to pMHC ligands of varying affinity might ultimately represent an overly simplified concept. Furthermore, it is likely that targets other than those affecting TCR signaling contribute to miR-181a/b-1 function in thymocyte subsets and peripheral T cells. The relative contribution of such targets in different contexts remains completely elusive. Specific deletion of miR-181 response elements in selected target genes might provide answers to these questions. Such experiments have been recently performed to analyze the miR-155/SOCS1 axis and the miR-17~92/Bim axis [93,94]. However, given that most likely co-targeting networks rather than individually targeted mRNAs constitute the dominant mode of action of miR-181, such experiments would require simultaneous mutation of multiple genes and might, therefore, be exceedingly difficult.

## Figures and Tables

**Figure 1 ijms-21-06200-f001:**
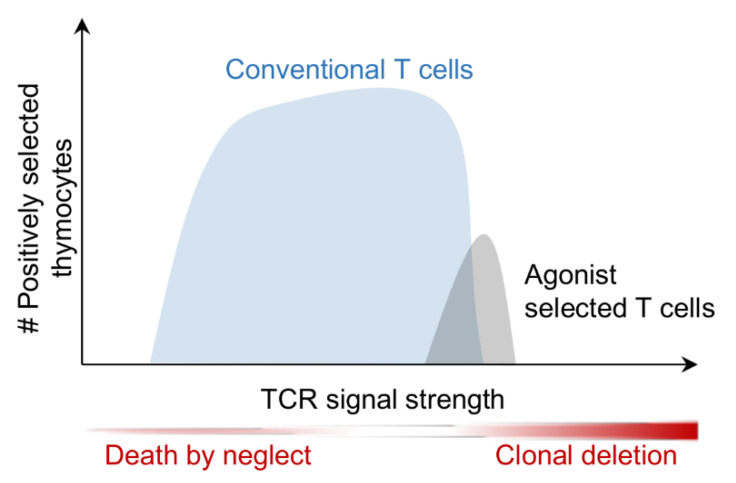
Conventional and agonist selected T cells possess different affinity windows for positive selection. Conventional T cells require low to moderate TCR affinities to peptide: MHC to receive survival signals, whereas agonist selected T cells depend on relatively high affinities towards peptide: MHC to be positively selected.

**Figure 2 ijms-21-06200-f002:**
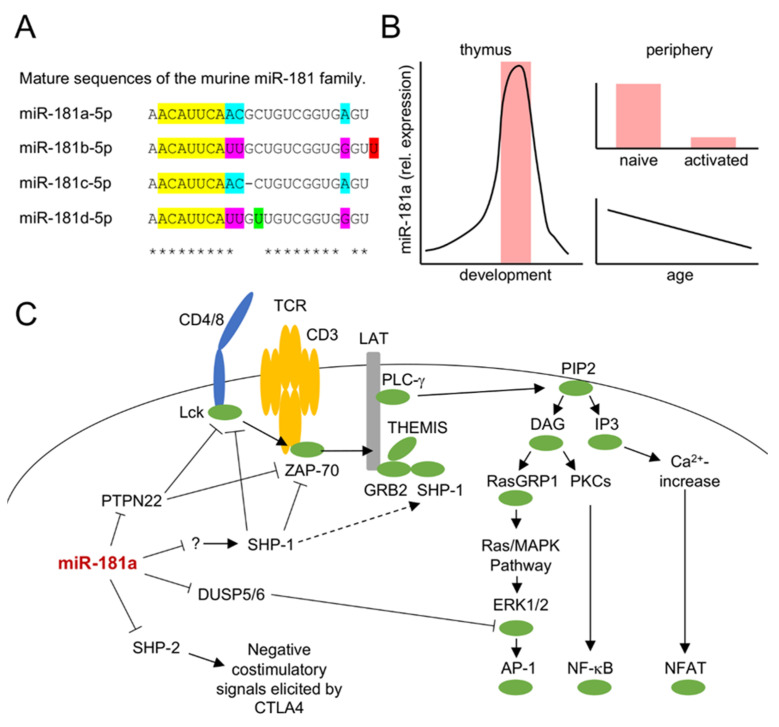
Characteristics of the miR-181 family. (**A**) Mature sequences of murine miR-181a, b, c, and d. Seed sequences are marked in yellow, asterisks indicate the consensus bases of all four microRNAs. Colored boxes highlight differences between family members. (**B**) Dynamic expression profiles of miR-181a(/b-1) in T cells from thymus (left) and periphery (right). The shaded area in the thymus indicates the phase of thymic selection. (**C**) Schematic depiction of miR-181a targets in the TCR signaling cascade. Positive and negative signaling relationships are indicated by solid lines. miR-181a targets PTPN22, DUSP5/6, and SHP-2, which, in turn, regulate Lck and ZAP-70, Erk1/2 and negative costimulatory signals elicited by CTLA4, respectively. Additionally, an indirect effect of miR-181a on SHP-1 has been proposed (denoted by question mark and dotted arrow).

**Figure 3 ijms-21-06200-f003:**
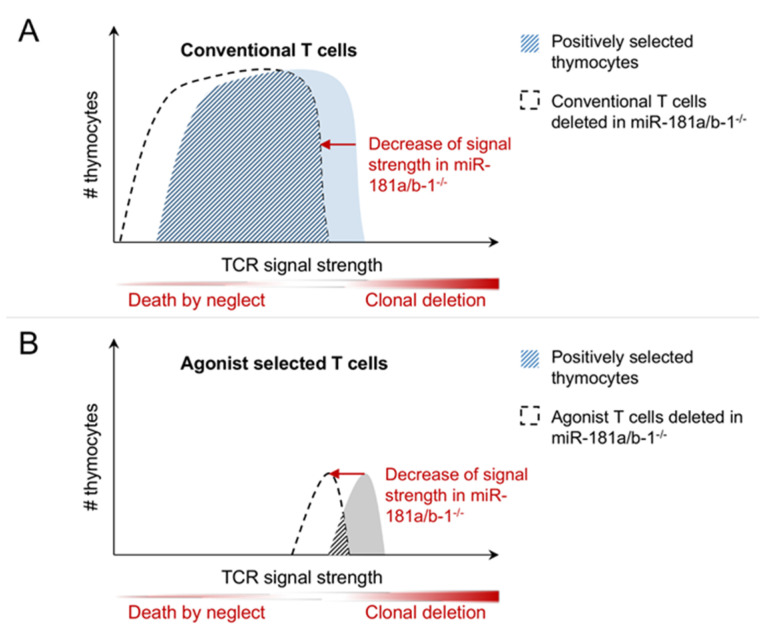
Hypothetical function of miR-181a/b-1 as rheostat for TCR signaling during selection. (**A**) How miR-181 affects conventional T cells: Due to their broad affinity window, the effect of the shift in TCR signal strength on selection is relatively small. (**B**) How miR-181 affects agonist selected T cells: Due to their high and narrow affinity window, the effect of the shift in TCR signal strength on selection is evident.

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
