# Peer review of "MicroRNA miR-181—A Rheostat for TCR Signaling in Thymic Selection and Peripheral T-Cell Function"

_ijms, 2020, doi:10.3390/ijms21176200_

Round 1
Reviewer 1 Report
The review article by Grewers and Krueger on micro-RNA 181 in TCR signaling, thymic selection and T cell function is a concise and yet comprehensive overview about the roles of this important micro-RNA in T cell biology. I have just a few comments that should be considered before publication.
Page 2, line 53/54: The authors might want to think about adding calcineurin to their list of important phosphatases regulating T cell signaling. After all, calcineurin is instrumental in the activation of NFAT, one of the three important transcription factor families leading to T cell activation (next to AP-1 and NF-kB; see Figure1).
Page 2, line 56: CD45 is not only a positive regulator. It can also de-phosphorylate TCR signaling molecules such as CD3zeta (see Courtney et al., Science Signal 2019 and references therein).
Page 4, lane 154: The authors state that the HY-TCR recognizes antigen in the context of MHC II. This is not correct. The HY-TCR recognizes the male (H-Y) antigen in the context of MHC class I H-2Db molecules.
Page 7, lane 264: The authors state that MAIT cells are mostly selected on thymocytes instead of epithelial cells. This is corrected. The authors might, however, think about mentioning here that MAIT cells are selected on antigens that are presented via the non-classical MHC molecule MR1 (Treiner et al., Nature 2003; Kjer-Nielsen et al., Nature 2012). After all, MR1 is mentioned in lane 277 without any further explanation.
Minor points:
Abstract, line ^13: “…family miR-181 family…” delete one family.
Page 1, line 30: in gdT cells gd should be Greek letters. The same applies to lanes 47 and 185.
Page 1, line 32: in abT cells ab should be Greek letters. The same applies to lanes 47 and 187.
Page 2, lane 87: “…function a s post-transcriptional…” should read “…function as post-transcriptional…”.
Page 3, lane 111: “… upon activation in in vitro,…” should read “… upon activation in vitro,…”.
Page 6, lane 210: TGFb – b should be a Greek letter.
Page 6, lane 218: “In order support…” should read “In order to support…”.
Page 6, lane 222: “miR-181a/b-1-defiecient” should read “miR-181a/b-1-deficient”.
Page 8, lane 315: “… model with conditional miR-181a/b-1 deficiency)” - delete bracket.
Page 8, lane 318/319: “…in this experimental was complex.” should read “…in this experimental system was complex.”
Author Response
Reviewer 1:
The review article by Grewers and Krueger on micro-RNA 181 in TCR signaling, thymic selection and T cell function is a concise and yet comprehensive overview about the roles of this important micro-RNA in T cell biology. I have just a few comments that should be considered before publication.
Page 2, line 53/54: The authors might want to think about adding calcineurin to their list of important phosphatases regulating T cell signaling. After all, calcineurin is instrumental in the activation of NFAT, one of the three important transcription factor families leading to T cell activation (next to AP-1 and NF-kB; see Figure1).
Response:
We have included calcineurin as phosphatase critical for TCR signaling (line 65).
Page 2, line 56: CD45 is not only a positive regulator. It can also de-phosphorylate TCR signaling molecules such as CD3zeta (see Courtney et al., Science Signal 2019 and references therein).
Response:
We have discussed the negative regulatory function of CD45 in line 63.
Page 4, lane 154: The authors state that the HY-TCR recognizes antigen in the context of MHC II. This is not correct. The HY-TCR recognizes the male (H-Y) antigen in the context of MHC class I H-2Db molecules.
Response:
We apologize for this obvious mistake, which has now been corrected.
Page 7, lane 264: The authors state that MAIT cells are mostly selected on thymocytes instead of epithelial cells. This is corrected. The authors might, however, think about mentioning here that MAIT cells are selected on antigens that are presented via the non-classical MHC molecule MR1 (Treiner et al., Nature 2003; Kjer-Nielsen et al., Nature 2012). After all, MR1 is mentioned in lane 277 without any further explanation.
Response:
We apologize for the confusion. MR1 and CD1d are first introduced in line 215. We have now added the relevant references, also for CD1d (Bendelac et al., Science 1995).
Minor points:
Abstract, line ^13: “…family miR-181 family…” delete one family.
Page 1, line 30: in gdT cells gd should be Greek letters. The same applies to lanes 47 and 185.
Page 1, line 32: in abT cells ab should be Greek letters. The same applies to lanes 47 and 187.
Page 2, lane 87: “…function a s post-transcriptional…” should read “…function as post-transcriptional…”.
Page 3, lane 111: “… upon activation in in vitro,…” should read “… upon activation in vitro,…”.
Page 6, lane 210: TGFb – b should be a Greek letter.
Page 6, lane 218: “In order support…” should read “In order to support…”.
Page 6, lane 222: “miR-181a/b-1-defiecient” should read “miR-181a/b-1-deficient”.
Page 8, lane 315: “… model with conditional miR-181a/b-1 deficiency)” - delete bracket.
Page 8, lane 318/319: “…in this experimental was complex.” should read “…in this experimental system was complex.”
Response:
We apologize for these formatting errors, which have now been corrected.
Reviewer 2 Report
In this review article, the authors summarized the previous progresses and studies about the miR-181 family in regulating the threshold of TCR signaling in T-cell selection during intra-thymic development. The authors presented an exhaustive review article on the involvement of the miR-181 family in conventional and unconventional T-cell differentiation, highlighting also the not yet clear roles and the discrepancies in literature data. Overall, the manuscript is well structured and contained novel and updated information. However, there are some minor revisions that the authors have to address to improve the quality of the manuscript:
1. Due to the large amount of information, some portions of the review are heavy and hard to follow. A table that summerize the dynamics of miR-181 expression in the development of specific t-cell subtypes, with the hypothetic mechanism of action (plus a row of references), would be helpful and could help readers to better follow the manuscript
2. Although authors cited appropriate references, some References are missing. E.g., from Line 23 to 38; line 102 should refere to reference [28]; line 129 should refere to [30]; line 325; and line 351 [80]
3. At line 95: before the introduction of the miR-181 family, could be nice a sentence/example of other relevant miRNAs involved in immune system development and function and why the authors focus the attention on miR-181 family
4. At line 101: the authors wrote “Mice lacking two of the three clusters have reduced body size and shortened overall survival, whereas deficiency of all three clusters appears to be incompatible with life, indicating that the miR-181 family exerts vital functions [27].”, than they describe the expression of the different clusters in T-cell lineage. However, the miR-181 family is also expressed in lungs, brain and eye, and is expressed in detectable levels in the bone marrow, spleen and muscles. The effect of their ablation on body size and survival could rely also on the multiple and pervasive roles of these miRNA family in different tissues in developing organism (10.3390/ijms21062092). The authors should mention these important aspects of miR-181 family roles in describing the above mentioned phenotype.
5. At line 102: although the miR-181c/d are very low expressed, some works reported a role of these two miRNA family members in T-cells (see for example: 10.1182/blood.V128.22.132.132 ; 10.1016/j.molimm.2010.10.021; 10.1371/journal.pone.0085274), the authors should at least mention these roles
6. From line 109 (but also later in the manuscript e.g. line 331): the authors begin to refere only to miR-181a roles. However, some of the work that they cite were carried out exploiting the ko mice of the entire miR-181a/b-1 cluster (see for example ref [31]). As also the authors mention, miR-181a and miR-181b present the same seed sequence and could recognize a similar set of targets. If the authors want to focus the attention only on miR-181a they should clearly claim and justify this decision or they should correct along the entire manuscript (and in Figure 1B) miR-181a in miR-181a/b when the description of the miR-181 role relies on studies carried out with a miR-181a/b-1 ko model.
7. Figure 2A is not cited at all in the text
8. The paragraph “miR-181 in unconventional T cells” (line 181-193) seems an introduction to the subsequent paragraphs (“Treg cells”; “iNKT cells and MAIT cells”). Indeed in this paragraph it is not described a role of miR-181 family members, that is indeed described in the subsequent paragraphs. To be more clear, I would like to suggest to the authors to number the paragraphs and sub-paragraphs of the manuscript
9. Finally, there are a few typos throughout the manuscript. For example:
- line 27
- line 32 and 47 (ab and gd should be αβ and γδ)
- line 74: the acronym for Double Positive (DP) should be introduced here and not at line 104
- line 87: there is a space between “a” and “s” of “as”
- line 89: the sentence “The miRNA-loaded RISC represses miRNA” should be “The miRNA-loaded RISC represses mRNA/target”
- line 248-249
- line 305
- line 315
- line 319
- line 286: fix the refereces (instead of [68] [69,70] should be [68-70]
- line 353: experimental autoimmune encephalitis is already referred as EAE at line 150
- line 353: fix the reference (Schaffert et al., 2015).
The authors should proof the manuscript.
10. In the “Founding” section the authors should add a statement about founding
Author Response
Reviewer 2:
Comments and Suggestions for Authors
In this review article, the authors summarized the previous progresses and studies about the miR-181 family in regulating the threshold of TCR signaling in T-cell selection during intra-thymic development. The authors presented an exhaustive review article on the involvement of the miR-181 family in conventional and unconventional T-cell differentiation, highlighting also the not yet clear roles and the discrepancies in literature data. Overall, the manuscript is well structured and contained novel and updated information. However, there are some minor revisions that the authors have to address to improve the quality of the manuscript:
- Due to the large amount of information, some portions of the review are heavy and hard to follow. A table that summerize the dynamics of miR-181 expression in the development of specific t-cell subtypes, with the hypothetic mechanism of action (plus a row of references), would be helpful and could help readers to better follow the manuscript
Response:
We have followed the reviewers advice and now provide a new Figure 2B summarizing dynamic expression of miR-181a(/b-1).
- Although authors cited appropriate references, some References are missing. E.g., from Line 23 to 38; line 102 should refere to reference [28]; line 129 should refere to [30]; line 325; and line 351 [80]
Response:
We thank the reviewer for the remark. Lines 23 to 38 constitute a general introduction into the topic at the level of an advanced textbook. Therefore, we decided not to overload this section with references. We have inserted references 28, 30 and 80 at the correct locations.
- At line 95: before the introduction of the miR-181 family, could be nice a sentence/example of other relevant miRNAs involved in immune system development and function and why the authors focus the attention on miR-181 family
Response:
We have included a number of miRNAs relevant for immune system development and function, included a number of references for comprehensive reviews and have outlined the focus of our manuscript in line 105.
- At line 101: the authors wrote “Mice lacking two of the three clusters have reduced body size and shortened overall survival, whereas deficiency of all three clusters appears to be incompatible with life, indicating that the miR-181 family exerts vital functions [27].”, than they describe the expression of the different clusters in T-cell lineage. However, the miR-181 family is also expressed in lungs, brain and eye, and is expressed in detectable levels in the bone marrow, spleen and muscles. The effect of their ablation on body size and survival could rely also on the multiple and pervasive roles of these miRNA family in different tissues in developing organism (10.3390/ijms21062092). The authors should mention these important aspects of miR-181 family roles in describing the above mentioned phenotype.
Response:
We have added the suggested review for descriptions of the variety of potential functions of miR-181 (line 115). Given the focused nature of our review, we have not expanded on that topic much further. To our knowledge, the lethal phenotype of pan-miR-181KO mice has not been analyzed in detail.
- At line 102: although the miR-181c/d are very low expressed, some works reported a role of these two miRNA family members in T-cells (see for example: 10.1182/blood.V128.22.132.132 ; 10.1016/j.molimm.2010.10.021; 10.1371/journal.pone.0085274), the authors should at least mention these roles
Response:
We have added a brief discussion of miR-181c/d (lines 118). Note that most of the published data rely on transgenic overexpression expression experiments and might therefore also be ascribed to miR-181a/b. Deletion of miR-181d alone had no discernable effects.
- From line 109 (but also later in the manuscript e.g. line 331): the authors begin to refere only to miR-181a roles. However, some of the work that they cite were carried out exploiting the ko mice of the entire miR-181a/b-1 cluster (see for example ref [31]). As also the authors mention, miR-181a and miR-181b present the same seed sequence and could recognize a similar set of targets. If the authors want to focus the attention only on miR-181a they should clearly claim and justify this decision or they should correct along the entire manuscript (and in Figure 1B) miR-181a in miR-181a/b when the description of the miR-181 role relies on studies carried out with a miR-181a/b-1 ko model.
Response:
The reviewer’s point is well taken and we have written an extensive addendum on this issue in line 123. We have also carefully checked our manuscript to distinguish between experiments performed in miR-181a/b-1 KO mice and overexpression or inhibition experiments generally referring to miR-181a alone. In particular, human studies to a large extent focus on miR-181a alone despite the fact that it is likely that miR-181b contributes to the observed findings.
- Figure 2A is not cited at all in the text
Response:
We have restructured our figures. Figure 2A is now Figure 1; Figure 2B,C is now Figure 3A,B.
- The paragraph “miR-181 in unconventional T cells” (line 181-193) seems an introduction to the subsequent paragraphs (“Treg cells”; “iNKT cells and MAIT cells”). Indeed in this paragraph it is not described a role of miR-181 family members, that is indeed described in the subsequent paragraphs. To be more clear, I would like to suggest to the authors to number the paragraphs and sub-paragraphs of the manuscript
Response:
We thank the reviewer for this suggestion and we have numbered the sections to highlight sub-sections.
- Finally, there are a few typos throughout the manuscript. For example:
- line 27
- line 32 and 47 (ab and gd should be αβ and γδ)
- line 74: the acronym for Double Positive (DP) should be introduced here and not at line 104
- line 87: there is a space between “a” and “s” of “as”
- line 89: the sentence “The miRNA-loaded RISC represses miRNA” should be “The miRNA-loaded RISC represses mRNA/target”
- line 248-249
- line 305
- line 315
- line 319
- line 286: fix the refereces (instead of [68] [69,70] should be [68-70]
- line 353: experimental autoimmune encephalitis is already referred as EAE at line 150
- line 353: fix the reference (Schaffert et al., 2015).
The authors should proof the manuscript.
Response:
We have carefully edited the manuscript.
- In the “Founding” section the authors should add a statement about founding
Response:
We have moved the funding section from the acknowledgments to the funding section.